# On the Limitation of Backdoor Detection Methods

**Georg Pichler**[1]    **Marco Romanelli**[2]
**Divya Prakash Manivannan**[2]    **Prashanth Krishnamurthy**[2]    **Farshad Khorrami**[2]
**Siddharth Garg**[2]
[1]TU Wien    [2]New York University

## Abstract

We introduce a formal statistical definition for the problem of backdoor detection in machine learning systems and use it to analyze the feasibility of such problem, providing evidence for the utility and applicability of our definition. The main contributions of this work are an impossibility result and an achievability result for backdoor detection. We show a no-free-lunch theorem, proving that universal backdoor detection is impossible, except for very small alphabet sizes. Furthermore, we link our definition to the probably approximately correct (PAC) learnability of the out-of-distribution detection problem, establishing a formal connection between backdoor and out-of-distribution detection.

## 1    Introduction

Safe and trustworthy Machine Learning (ML) systems remain elusive [8, 15], for reasons that are intrinsic, like poor interpretability [1, 13], and due to external threats, including inference time adversarial inputs [6, 8, 4] and training time poisoning and backdoor attacks [5]. As the scale, complexity and training data requirements of modern deep neural network architectures has grown, many users resort to using and/or fine-tuning pre-trained models. Consequently, purposefully implanted backdoors pose a security risk for ML systems.

In the classic backdoor threat model, a malicious actor may provide poisoned data, affecting the behavior of trained ML model. For certain, *poisoned* inputs, which are modified in a specific way, known to the attacker, the model then provides erroneous predictions. While there are many ways such a backdoor could be embedded into a model, prior work shows that poisoning even a small fraction of training data yields models with stealthy and effective backdoors [12]. To detect a backdoor, the model user (i.e., the *defender*) has access to a, typically small, validation dataset of clean inputs. In the Model Backdoor Detection (MBD) problem [9, 7], the defender wishes to detect if the model itself contains a backdoor. In the Input Backdoor Detection (IBD) problem [10, 11], the defender wants to test if a specific test input is poisoned or not. Yet, despite several years of research, the field is still plagued by the cat-and-mouse game between attacks and defenses, with no end in sight. Unlike the work on adversarial perturbation attacks, for instance, "certifiable" defenses have remained elusive. We argue that this is in part because, despite the large body of work in the area, backdoor detection has not been formally defined, at least not in a precise and well-posed manner. This lack of formal treatment has negative consequences as it impedes fair and consistent comparison between methods.

**Contributions.** In this paper, we present the first precise statistical formulation of the MBD and IBD problems (Section 2.1). This formulation enables several new insights on backdoor detection. (1) *Relationship to well-known statistical problems:* Our formulation unifies MBD, IBD and even Out-Of-Distribution (OOD) detection within a common framework and we reduce these problems to standard statistical hypothesis testing problems. (2) *Infeasibility:* Leveraging these reductions, we conclude that under realistic assumptions, universal (adversary-unaware) backdoor detection is not possible for an infinite alphabet of the training data. (3) *Bound for finite alphabet size:* For a finite

Published at NeurIPS 2023 Workshop on Backdoors in Deep Learning: The Good, the Bad, and the Ugly.

data alphabet, we provide a bound on the achievable error probability given a fixed training set size. These bounds are evaluated for commonly used datasets in ML, showing that universal backdoor detection is only achievable for very small alphabets. (4) *Connections to Probably Asymptotically Correct (PAC) learning theory of OOD detection:* Detecting a backdoor in training data is equivalent to a binary Neyman-Pearson hypothesis test if OOD detection is PAC learnable as defined in [3].

## 2 Theoretical Formulation and Results

We focus on MBD and IBD in the case where the attacker has limited control over the training data and is able to poison a certain portion of the dataset. The training itself is performed using a standard method, e.g., Stochastic Gradient Descent (SGD). For an extensive overview of other empirical backdoor problems, the reader is referred to, e.g., [15].

### 2.1 Formulating Model Backdoor Detection (MBD)

**Overview.** After $N$ samples of training data are collected, the backdoor attacker has the option of poisoning a portion of the training data, by replacing each *clean* sample with a *poisoned* sample. This poisoning may alter, e.g., images as well as their labels. Subsequently, an Artificial Neural Network (ANN) is trained on the resulting training set. Given the resulting trained network (i.e., the network parameters), the task of the backdoor detector is to determine whether the training data had been poisoned. The detector may obtain $M$ additional *clean* samples, e.g., by independently collecting additional data. We assume that the backdoor attacker has no access to these samples.

**Dataset and training.** Consider a, possibly stochastic, training algorithm $\mathcal{A}$ (e.g., SGD), that trains a model on training data[1] $\mathcal{D}_N = (X_1, X_2, \ldots, X_N)$, consisting of $N$ i.i.d. random variables, distributed like $X \sim P$, as input and produces a parameter vector $\theta = \mathcal{A}(\mathcal{D}_N)$ as output.

**Clean data.** Let $P_0 \in \mathcal{P}(\mathcal{X})$ be the probability distribution on $\mathcal{X}$ of clean samples and let $\mathcal{D}_N^{(0)} = (X_1^{(0)}, X_2^{(0)}, \ldots, X_N^{(0)})$ be a clean dataset, consisting of $N$ i.i.d. random variables, drawn from $P_0$.

**Backdoor.** To *backdoor* a model trained on the clean dataset, an adversary may replace some training samples with poisoned samples drawn from a different distribution $P_b \in \mathcal{P}(\mathcal{X})$. As a training sample may include the data and the label, the adversary could change the data and label.

**Poisoned training data.** Assuming that a fraction $\gamma \in (0, 1]$ of the training data is poisoned, the poisoned training dataset $\mathcal{D}_N^{(1)} = (X_1^{(1)}, X_2^{(1)}, \ldots, X_N^{(1)})$ is independently drawn according to $P_1 = \gamma P_b + (1 - \gamma)P_0$, i.e., according to $P_b$ with probability $\gamma$ and from $P_0$ with probability $1 - \gamma$.

**Additional clean data.** Furthermore, let $\mathcal{D}'_M = (X'_1, X'_2, \ldots, X'_M) \sim P_0^M$ be $M$ i.i.d. additional clean samples distributed according to $P_0$. These samples correspond to clean validation data or may have been collected by the backdoor detector prior to making a decision.

**Model Backdoor Detection.** The backdoor detector is a function $g$, that takes $\theta = \mathcal{A}(\mathcal{D}_N^{(j)})$ and additional data $\mathcal{D}'_M$ as its input and outputs 0 for *"backdoor"* and 1 for *"no backdoor"*. For MBD, we require the detector to determine $j$ with high probability. For ease of notation, we use a Bernoulli-$\frac{1}{2}$ random variable $J \sim \mathcal{B}(\frac{1}{2})$ and define the input for the detector as $\mathbf{Q} = (\mathcal{A}(\mathcal{D}_N^{(J)}), \mathcal{D}'_M)$, such that the error probability $\Pr\{g(\mathbf{Q}) \neq J\}$ of the detector is well-defined.

**Possible data distributions.** The last observation to obtain a well-defined backdoor detection problem is that we need to avoid the possibility of $P_0 = P_b$. Detection is impossible if the clean and the backdoor distributions are identical. We opt for the general approach of defining a suitable set $\mathcal{P} \subseteq \mathcal{P}(\mathcal{X})^2$ that contains all possible clean and backdoor distribution pairs $(P_0, P_b) \in \mathcal{P}$.

These discussions then naturally lead to the following central definition.

**Definition 1.** *The MBD problem for a training algorithm $\mathcal{A}$ is determined by the following quantities:* $\gamma \in (0, 1]$, $N \in \mathbb{N}$, $M \in \mathbb{N}$, *and* $\mathcal{P} \subseteq \mathcal{P}(\mathcal{X})^2$.

---

[1]The training sample $X$ may be a vector that includes data and label.

*Fixing these quantities, we define the* risk *of a backdoor detector g associated with* $(P_0, P_b)$ *as*

$$R(g; P_0, P_b) := \Pr\{g(\mathbf{Q}) \neq J\} = \frac{1}{2} \sum_{j=0,1} \Pr\{g(\mathcal{A}(\mathcal{D}_N^{(j)}), \mathcal{D}'_M) \neq j\}. \tag{1}$$

*We say that a backdoor detector is* $\alpha$*-error for some* $\alpha \in [0, \frac{1}{2}]$ *if, for every pair* $(P_0, P_b) \in \mathcal{P}$*, the risk is bounded by*

$$R(g; P_0, P_b) \leq \alpha. \tag{2}$$

*Remark* 1. Instead of bounding the risk as in (2), it may seem more natural to require $\Pr\{g(\mathbf{Q}) \neq j | J = j\} \leq \alpha$ for both $j = 0, 1$, but note that $\Pr\{g(\mathbf{Q}) \neq j | J = j\} \leq 2\alpha$ for $j = 0, 1$ immediately follows from (2).

## 2.2 (In)feasibility of Model Backdoor Detection

It will be useful to consider easier problems than $\alpha$-error detection, as defined in Definition 1 and establish reductions. To this end, we consider four different *Types* of detectors. All these detectors need to infer $J$, but different inputs are available to them:

0. $g_0(\mathbf{Q}_0)$ with $\mathbf{Q}_0 = \mathbf{Q} = (\mathcal{A}(\mathcal{D}_N^{(J)}), \mathcal{D}'_M)$: The default detector as used in Definition 1.

1. $g_1(\mathbf{Q}_1)$ with $\mathbf{Q}_1 = (\mathcal{D}_N^{(J)}, \mathcal{D}'_M)$: Provide the detector with the training dataset $\mathcal{D}_N^{(J)}$ and $M$ independent clean samples $\mathcal{D}'_M$.

2. $g_2(\mathbf{Q}_2)$ with $\mathbf{Q}_2 = (\mathcal{D}_N^{(J)}, P_0)$: Provide the detector with the training dataset $\mathcal{D}_N^{(J)}$, and with the clean data distribution $P_0$.

3. $g_3(\mathbf{Q}_3)$ with $\mathbf{Q}_3 = (\mathcal{D}_N^{(J)}, P_0, P_b)$: Provide the detector with the training dataset $\mathcal{D}_N^{(J)}$, the clean distribution $P_0$ and with the backdoor distribution $P_b$. This is a binary Neyman-Pearson hypothesis testing problem between $P_0^N$ and $P_1^N$.

We assume that detectors of Types 2 and 3 have access to $P_0$ (and $P_b$ for a Type 3 detector) in terms of evaluation of the distribution, and also have the ability to sample from the distribution. We thus consider Types 2 and 3 as randomized detectors to account for sampling. The definitions of risk and $\alpha$-error detection of $g_2, g_3$ apply mutatis mutandis as in Definition 1, where the probability in (1) is also taken over the randomness of $g$.

Remark 3 proposes an ordering of the Types of detectors, according to the information provided.

In Section 2.2.1 we will show that for a reasonable $\mathcal{P}$, $\alpha$-error Type 2 detection is impossible with $\alpha < \frac{1}{2}$. The reduction argument in Remark 3 thus ensures that $\alpha$-error detection with $\alpha < \frac{1}{2}$ is also impossible for Type 0 and Type 1 detectors.

We can resolve the situation for a Type 3 detector using the Neyman-Pearson lemma.

**Lemma 1.** *Given a Type 3 backdoor detector* $g_3(\mathcal{D}_N, P_0, P_b)$*, for any pair* $(P_0, P_b) \in \mathcal{P}(\mathcal{X})^2$

$$R(g_3; P_0, P_b) \geq \frac{1}{2} - \frac{1}{2} \mathrm{TV}(P_0^N, P_1^N) \geq \frac{1}{2} - \frac{\gamma N}{2} \mathrm{TV}(P_0, P_b), \tag{3}$$

*where the first equality in* (3) *can be achieved by the Neyman-Pearson detector. Thus, an* $\alpha$*-error detector of Type 3 can only exist if* $\alpha \geq \frac{1}{2} - \frac{\gamma N}{2} \mathrm{TV}(P_0, P_b)$ *for all* $(P_0, P_b) \in \mathcal{P}$.

See proof in Appendix A.3 .

Before analyzing Types 1 and 2, we specify the set of allowable distributions $\mathcal{P}$ using Lemma 1.

Merely excluding the identity $P_0 \neq P_b$, i.e., $\mathcal{P} = \{(P_0, P_b) \in \mathcal{P}(\mathcal{X})^2 : P_0 \neq P_b\}$ is not sufficient. *Example* 1. Let $g_3(\mathcal{D}_N, P_0, P_1)$ be an $\alpha$-error Type 3 detector and assume that $\mathcal{X}$ is infinite, i.e., $|\mathcal{X}| = \infty$. Let $\mathcal{P}$ be given as above, ensuring only that $P_0 \neq P_b$. For any $\varepsilon > 0$, we can then choose[2] $(P_0, P_b) \in \mathcal{P}$ with $0 < \mathrm{TV}(P_0, P_b) \leq \frac{2}{\gamma N}\varepsilon$. By Lemma 1, we have $\alpha \geq \frac{1}{2} - \frac{\gamma N}{2} \mathrm{TV}(P_0, P_b) \geq \frac{1}{2} - \varepsilon$. As $\varepsilon > 0$ was arbitrary, we have $\alpha = \frac{1}{2}$.

---

[2]Without loss of generality, we can assume $\mathcal{X} = \mathbb{N}$. Then, this can, e.g., be achieved by $P_0 = \mathcal{U}(\{0, 1, 2, \ldots, \lfloor \frac{\gamma N}{2\varepsilon} \rfloor\})$ and $P_1 = \mathcal{U}(\{1, 2, \ldots, \lfloor \frac{\gamma N}{2\varepsilon} \rfloor\})$. We use $\mathcal{U}(\cdot)$ to denote a uniform distribution on a finite set.

Lemma 1 and Example 1 show that even for a Type 3 detector, we need $\text{TV}(P_0, P_b) > \frac{1-2\alpha}{\gamma N}$ for all $(P_0, P_b) \in \mathcal{P}$, in order for $\alpha$-error detection to be achievable. In the following we will assume that $\mathcal{P}$ is the set of probability distributions $P_0, P_b$ with $\text{TV}(P_0, P_b) \geq 1 - \beta$, for some fixed $\beta \in [0, 1)$. This strong requirement is motivated by the fact that in this case, $\frac{1-\gamma+\gamma\beta}{2}$-error Type 3 detection is achievable with only $N = 1$ sample.

*Remark* 2. Thorough reasoning and examples, illustrating why total variation distance is the preferred distance measure for distribution hypothesis testing can be found in [2, Section 1.2].

### 2.2.1 Impossibility

In the following we prove an impossibility result, which implies that *for an infinite alphabet $\mathcal{X}$, the error probability (as given in Definition 1) of any detector (of Type 0, Type 1 or Type 2) is $\frac{1}{2}$, the error probability of a random guess.* Additionally, for finite $\mathcal{X}$, we provide a lower bound on the size of the training set $N$, as a function of $\alpha$.

**Theorem 1.** *Fix $N \in \mathbb{N}$, $\alpha \in (0, \frac{1}{2}]$, $\beta \in [0, 1]$, and $\mathcal{P} = \{(P_0, P_b) : \text{TV}(P_0, P_b) \geq 1 - \beta\}$. Let $g_2(\mathcal{D}_N, P_0)$ be an $\alpha$-error Type 2 detector. For $|\mathcal{X}| = \infty$, we then have necessarily $\alpha = \frac{1}{2}$, while for $|\mathcal{X}| < \infty$, we have*

$$N \geq \frac{\log 2\alpha}{2} + \sqrt{\frac{(\log 2\alpha)^2}{4} + (\beta|\mathcal{X}| - 1)\log\frac{1}{2\alpha}}. \tag{4}$$

See proof in Appendix A.3 .

For a fixed dataset alphabet size $|\mathcal{X}|$ and allowed error probability $\alpha$, the bound (4) gives the minimum size of the training set $N$ for the error level $\alpha$ to be achievable. Note the following special cases in terms of $\alpha$, $\beta$: *i)* For $\alpha = \frac{1}{2}$, the bound (4) is always satisfied as the RHS is 0, showing that $\frac{1}{2}$-error detection is always achievable. This coincides with the error probability of a random guess. *ii)* The bound (4) is monotonically decreasing in $\alpha$ and for $\alpha \to 0$, it approaches $\beta|\mathcal{X}|$. *iii)* In case $\beta = 0$, the bound (4) is always satisfied as the RHS is zero for $\alpha \in (0, \frac{1}{2}]$ in this case. This shows that $\alpha$-error detection is always possible if $P_0$ and $P_b$ have disjoint support, i.e. $\text{TV}(P_0, P_b) = 1$.

For an infinite alphabet $\mathcal{X}$, (4) needs to be satisfied for arbitrarily large values of $|\mathcal{X}|$. For finite training set size $N$, this is only possible if $\alpha = \frac{1}{2}$ as then, $\log\frac{1}{2\alpha} = 0$. Thus, in this case, for any Type 2 detector, there is a particular clean distribution and backdoor strategy, such that this detector performs no better than random guessing. For fixed $\alpha$ and $\beta$, we can use (4) to determine the minimum size of the training set $N$ for popular datasets, for $\alpha$ error probability to be achievable by a Type 2 detector. To this end, we use the width $W$, height $H$, number of channels $C$ and color depth $P$ of an image dataset to compute $|\mathcal{X}| = P^{WHC}$. For categorical datasets, we may multiply the number of categories for all the properties recorded in the dataset to obtain $|\mathcal{X}|$. The resulting value for the bound in (4) is given in Table 1 for several popular datasets. As can be seen by these numbers, this universal backdoor detection is infeasible for all, but the smallest tabular datasets. Note also, that the impossibility of Type 2 backdoor detection automatically precludes the existence of Type 1 or Type 0 error detectors with equal performance by the reduction argument in Remark 3.

### 2.2.2 Achievability

In this section we are going to show that achievability is possible and that it is related to the size of the alphabet $|\mathcal{X}|$. We consider a Type 2 detector and give a criterion for $\alpha$-error detection achievability:

**Theorem 2.** *Considering the backdoor detection setup of Definition 1 with $\mathcal{P} = \{(P_0, P_b) : \text{TV}(P_0, P_b) \geq 1 - \beta\}$ and a finite alphabet $|\mathcal{X}| < \infty$. There exists an $\alpha$-error Type 2 detector if*

$$\alpha > 2|\mathcal{X}|\exp\left(-\frac{2N\gamma^2(1-\beta)^2}{|\mathcal{X}|^2}\right), \tag{5}$$

See proof in Appendix A.3 .

Note the following special cases in terms of $\alpha$, $\beta$ and $\gamma$: *i)* For $\alpha = 0$, (5) cannot be satisfied, showing that 0-error detection cannot be achieved. *ii)* The case $\beta = 1$ allows for $P_0 = P_b$ and thus

no $\alpha$-error detector exists for $\alpha \in [0, \frac{1}{2})$ in this case and (5) cannot be satisfied. *iii)* For $\gamma = 1$, $P_0 = P_b$ are identical, no $\alpha$-error detector exists for $\alpha \in [0, \frac{1}{2})$, and (5) cannot be satisfied.

### 2.2.3 Connections to PAC-Learnability of OOD Detection

Note that a Type 1 detector essentially needs to solve an OOD detection problem. In this case, we are provided samples and the detector $g_1$ needs to determine if the $N$ samples $\mathcal{D}_N$ were drawn from the same distribution as $\mathcal{D}'_M$.

The goal of this section is to prove Theorem 3. This theorem has an interesting implication in case the OOD detection problem is PAC-learnable: If an $\alpha$-error Type 3 backdoor detector $g_3$ exists, then $(\alpha + \epsilon)$-error detection is also possible for a Type 1 detector for any $\epsilon > 0$. Thus, essentially Types 1 to 3 all become equivalent if OOD detection is PAC-learnable. Note here that Type 3 detection is completely characterized by Lemma 1.

Table 1: Lower bound (4) on $N$ evaluated for popular datasets with $\alpha = 0.1$ and $\beta = 0.001$.

| Dataset | $|\mathcal{X}|$ | N |
|---|---|---|
| Lisa Traffic Sign | $256^{307200}$ | $\geq 10^{369904}$ |
| ImageNet | $256^{150528}$ | $\geq 10^{181252}$ |
| CIFAR10 | $256^{3072}$ | $\geq 10^{3697}$ |
| MNIST | $256^{784}$ | $\geq 10^{942}$ |
| B/W MNIST | $2^{784}$ | $\geq 10^{116}$ |
| Adult | $\geq 10^{21.86}$ | $\geq 10^9$ |
| Heart Disease | $\geq 10^{13.51}$ | $\geq 10^5$ |
| Iris | $\geq 10^{6.35}$ | $\geq 10^1$ |

The PAC-learnability of the detector in Type 1 was analyzed in [3]. We fist restate a special case of the definition of (weak) PAC-learnability as given in [3, Def. 1].

**Definition 2.** *For distributions $P_0, P_b$ on $\mathcal{X}$, the* OOD-risk *of a function $f : \mathcal{X} \to \{0, 1\}$, w.r.t. the Hamming distance, is defined as*

$$\bar{R}(f, P_0, P_b) := \Pr\{f(X^{(J)}) \neq J\} = \frac{1}{2} \Pr\{f(X^{(0)}) = 1\} + \frac{1}{2} \Pr\{f(X^{(1)}) = 0\}. \quad (6)$$

*Given a space of probability function $\mathcal{P}$,* OOD-detection *is PAC-learnable on $\mathcal{P}$ if there exists an algorithm $\mathcal{G} \colon \bigcup_{m=1}^{\infty} \mathcal{X}^m \to \{0, 1\}^{\mathcal{X}}$ and a monotonically decreasing sequence $\epsilon(m)$ such that $\lim_{m \to \infty} \epsilon(m) = 0$ and for all $(P_0, P_b) \in \mathcal{P}$, and all $m \in \mathbb{N}$ we have*

$$\mathbb{E}[\bar{R}(\mathcal{G}(\mathcal{D}'_m), P_0, P_b)] - \inf_f \bar{R}(f, P_0, P_b) \leq \epsilon(m), \quad (7)$$

*where the expectation is taken w.r.t. $\mathcal{D}'_m$ and the infimum is over $\{0, 1\}^{\mathcal{X}}$, i.e., all $f \colon \mathcal{X} \to \{0, 1\}$.*

Appendix A.2 shows how Definition 2 is a special case of [3, Def. 1]. We consider PAC-learnability on the $N$-dimensional product space, i.e., on $\mathcal{X}^N$ with distributions $P_0^N, P_b^N$. We can now connect PAC-learnability to the existence of $\alpha$-error detectors of Types 1 and 3.

**Theorem 3.** *Consider the setup of Definition 1, with fixed $\gamma \in (0, 1]$, $N \in \mathbb{N}$ and $\mathcal{P}$. Let $\mathcal{P}'$ be the set of $N$-fold products of $(P_0, P_1)$, i.e., $\mathcal{P}' = \{(P_0^N, (\gamma P_b + (1 - \gamma)P_0)^N) : (P_0, P_b) \in \mathcal{P}\}$. Then, OOD-detection is PAC-learnable on $\mathcal{P}'$ if and only if the following holds for any $\epsilon > 0$ and any Type 3 detector $g_3(\mathcal{D}_N, P_0, P_b)$: We can find $M \in \mathbb{N}$ and a Type 1 detector $g_1(\mathcal{D}_N, \mathcal{D}'_M)$, which satisfies $R(g_1, P_0, P_b) \leq R(g_3, P_0, P_b) + \epsilon$ for every $(P_0, P_b) \in \mathcal{P}$.*

See proof in Appendix A.3 .

**Corollary 1.** *If OOD-detection is PAC-learnable on $\mathcal{P}'$, we have the following: If $\alpha$-error backdoor detection is possible in the easier case of Type 3 detection, which is completely characterized by Lemma 1, then $(\alpha + \epsilon)$-error detection is also possible for a Type 1 detector for any $\epsilon > 0$. Consequently, the achievability of backdoor detection for Types 1 to 3 detectors are all equivalent up to topological closure if OOD-detection is PAC-learnable on $\mathcal{P}'$.*

### 2.3 Generalizing to Input Backdoor Detection

We generalize Definition 1 to IBD. Let $g'(\mathbf{Q}')$ take input $\mathbf{Q}' = (\mathbf{Q}, X^{(I)}) = (\mathcal{A}(\mathcal{D}_N^{(J)}), \mathcal{D}'_M, X^{(I)})$, where a random variable $I$ on $\{0, 1\}$ determines if $X^{(I)}$ was drawn as $X^{(0)} \sim P_0$ ($I = 0$) or as[3] $X^{(1)} \sim P_b$ ($I = 1$). We define a general target function $t(j, i) \in \{0, 1\}$ and require that a backdoor detector satisfies $g'(\mathbf{Q}') = t(J, I)$ with high probability. In this case, it is beneficial to allow for an arbitrary probability distribution $P_{JI}$ of $(J, I)$ on $\{0, 1\}^2$. This leads to the following definition

---

[3]Note, that $X^{(1)}$ is distributed according to $P_b$ and **not** according to $P_1 = (1 - \gamma)P_0 + \gamma P_b$.

**Definition 3.** *A backdoor detection problem for a training algorithm $\mathcal{A}$ is determined by the following quantities: $\gamma \in (0,1]$, $N \in \mathbb{N}$, $M \in \mathbb{N}$, $\mathcal{P} \subseteq \mathcal{P}(\mathcal{X})^2$, $P_{JI} \in \mathcal{P}(\{0,1\}^2)$, and $t\colon \{0,1\}^2 \to \{0,1\}$. Fixing these quantities, we define the* risk *of a backdoor detector $g'$ associated with $(P_0, P_b)$ as $R(g'; P_0, P_b) := \Pr\{g'(\mathbf{Q}') \neq t(J, I)\}$, where the probability is w.r.t. $\mathbf{Q}' = (\mathcal{A}(\mathcal{D}_N^{(J)}), \mathcal{D}_M', X^{(I)})$ and $(J, I) \sim P_{JI}$. We say that a backdoor detector is $\alpha$-error for some $\alpha \in [0, \frac{1}{2}]$ if, for every pair $(P_0, P_b) \in \mathcal{P}$, the risk is bounded by $R(g'; P_0, P_b) \leq \alpha$.*

OOD can be modeled using the target function $t(j, i)$ for MBD, IBD and OOD Figure 1.

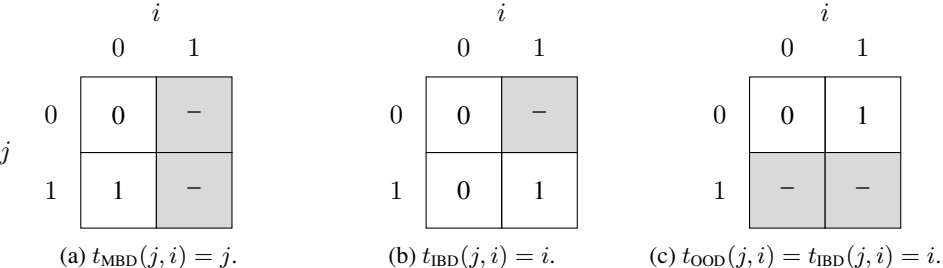

(a) $t_{\text{MBD}}(j, i) = j$.  (b) $t_{\text{IBD}}(j, i) = i$.  (c) $t_{\text{OOD}}(j, i) = t_{\text{IBD}}(j, i) = i$.

Figure 1: Target function $t(j, i)$ for different backdoor detection flavors.

Note that several cells in the diagrams in Fig. 1 are grayed out. This reflects the fact that for certain flavors of backdoor detection, specific combinations of $(j, i)$ are not relevant. For MBD for instance, we are not interested in whether the target sample $X^{(I)}$ contains a backdoor and we can thus assume $I = 0$ in this case, effectively reducing this case to the problem introduced in Section 2.1 with $M+1$ samples being drawn from $P_0$, i.e., $(\mathcal{D}_M', X^{(0)}) = \mathcal{D}_{M+1}'$, available to the detector. Conversely, the case of a clean model, i.e., $j = 0$ and a sample with a backdoor, i.e., $i = 1$ is not realistic for IBD and we set $P_{JI}(0, 1) = 0$ in this case. By setting $J = 0$ (i.e., model is trained on clean data and $P_{JI}(1, 0) = P_{JI}(1, 1) = 0$) and using $t_{\text{OOD}}(j, i) = t_{\text{IBD}}(i, 0) = i$, we obtain an OOD detection problem, where the detector has access to a model $\mathcal{A}(\mathcal{D}_N^{(0)})$ trained on clean data and additional clean data $\mathcal{D}_M'$. The detector then needs to determine whether $X^{(I)}$ is in-distribution ($I = 0$) or out-of-distribution ($J = 1$). To showcase, how our result from Sections 2.2.1 and 2.2.2 carry over to other variants of backdoor detection, we will directly use Theorem 1 to derive a similar result for IBD. In analogy to the different Types of MBD detectors introduced in Section 2, we have a Type 2 detector $g_2'(\mathbf{Q}_2')$ with $\mathbf{Q}_2' = (\mathcal{D}_{N'}^{(J)}, P_0, X^{(I)})$ for IBD. For such a detector we can leverage a reduction argument to obtain the following.

**Corollary 2.** *Let $g_2'(\mathcal{D}_{N'}^{(J)}, P_0, X^{(I)})$ be a Type 2 detector for an IBD problem with $r = \min\{P_{JI}(0, 0), P_{JI}(1, 1)\} > 0$ and $\mathcal{P} = \{(P_0, P_b) : \text{TV}(P_0, P_b) \geq 1 - \beta\}$. Then, if $g_2'$ is $\alpha$-error, we have $\alpha \geq r$ if $|\mathcal{X}| = \infty$, and for $|\mathcal{X}| < \infty$, we obtain*

$$N \geq \frac{\log \frac{\alpha}{r}}{2} + \sqrt{\frac{(\log \frac{\alpha}{r})^2}{4} + (\beta|\mathcal{X}| - 1)\log \frac{r}{\alpha}}. \tag{8}$$

See proof in Appendix A.3 .

## 3 Conclusions

We provided a formal statistical definition of backdoor detection and investigated the feasibility of backdoor detection. We concluded that under realistic assumptions, universal (adversary-unaware) backdoor detection is not possible. Thus, effective backdoor detectors need to be adversary-aware.

## Acknowledgement

This work is supported in part by grants from the National Science Foundation (NSF) and the ARO 77191, in part by the Army Research Office under grant #W911NF-21-1-0155, and in part by the NYUAD Center for Artificial Intelligence and Robotics, funded by Tamkeen under the NYUAD Research Institute Award CG010.

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

# A Appendix

## A.1 Ordering of detector Types

*Remark* 3 (Ordering of detector Types). Types 0 to 3 are listed in order of decreasing difficulty as, e.g., more information is provided to a Type 3 detector than to a Type 2 detector. Thus, an $\alpha$-error detector $g$ immediately provides an $\alpha$-error Type 1 detector $g_1$, which in turn immediately provides an $\alpha$-error Type 2 detector $g_2$, which yields an $\alpha$-error detector $g_3$ of Type 3. Thus, we can define a total ordering on the different Types of detectors, using $A \prec B$ to signify that $A$ can be derived from $B$: $\mathbf{Q}_0 \prec \mathbf{Q}_1 \prec \mathbf{Q}_2 \prec \mathbf{Q}_3$. The formal argument, showing this claim can be found in Lemma 2.

## A.2 Definition 2 is a special case of [3, Def. 1]

*Remark* 4. Definition 2 is a special case of [3, Def. 1] in several ways[4]: *i)* The hypothesis space is the complete function space $\mathcal{H} = \{0,1\}^{\mathcal{X}}$, of functions $f \colon \mathcal{X} \to \{0,1\}$. *ii)* The loss function, as used in [3, Eq. (1)] is the Hamming distance, i.e., $\ell(y,y') = 1$ if and only if $y \neq y'$. *iii)* We are purely concerned with one-class novelty detection, i.e., $K = 1$ in [3, Sec. 2]. Therefore we do not take $Y_O$ and $Y_I$ into account, as $Y_I \equiv 1$ and $Y_O \equiv 2$. *iv)* Note that $(P_0, P_b) \in \mathcal{P}$ play the role of $(D_{X_O}, D_{X_I})$ and the complete domain space is then given by $\mathscr{D}_{XY} = \{D_{XY} : D_{XY} = \frac{1}{2}P_0 + \frac{1}{2}P_b, (P_0, P_b) \in \mathcal{P}\}$. Besides, strong PAC-learnability [3, Def. 2] implies weak learnability.

## A.3 Proofs

**Lemma 2.** *Let $g_l$ be a detector as listed in Section 2 with input $\mathbf{Q}_l$ for $l \in \{0,1,2\}$, where we set $g_0 = g$ and $\mathbf{Q}_0 = \mathbf{Q}$. If $g_l$ is $\alpha$-error in the sense of Definition 1, then for $m \in \{1,2,3\}$ and $m > l$ we can find a backdoor detector $g_m$ with input $\mathbf{Q}_m$ that is also $\alpha$-error.*

*Proof of Lemma 2.* It is sufficient to show the lemma for $m = l + 1$. The claim then follows by applying the result repeatedly.

In the case $l = 2$ (and $m = 3$) we obtain $g_3$ with $R(g_3, P_0, P_b) = R(g_2, P_0, P_b)$ by $g_3(\mathcal{D}, P_0, P_b) = g_2(\mathcal{D}, P_0)$.

For $l = 1$, we can define the randomized detector $g_2(\mathcal{D}, x, P_0)$ to first draw $M$ i.i.d. samples $\mathcal{D}'_M \sim P_0^M$ and then yield $g_2(\mathcal{D}, P_0) = g_1(\mathcal{D}, \mathcal{D}'_M)$.

Finally, for $l = 0$ we obtain $g_1$ with equal risk by defining $g_1(\mathcal{D}, \mathcal{D}') = g(\mathcal{A}(\mathcal{D}), \mathcal{D}')$.

$\square$

**Lemma 1.** *Given a Type 3 backdoor detector $g_3(\mathcal{D}_N, P_0, P_b)$, for any pair $(P_0, P_b) \in \mathcal{P}(\mathcal{X})^2$*

$$R(g_3; P_0, P_b) \geq \frac{1}{2} - \frac{1}{2}\,\mathrm{TV}(P_0^N, P_1^N) \geq \frac{1}{2} - \frac{\gamma N}{2}\,\mathrm{TV}(P_0, P_b), \tag{3}$$

*where the first equality in (3) can be achieved by the Neyman-Pearson detector. Thus, an $\alpha$-error detector of Type 3 can only exist if $\alpha \geq \frac{1}{2} - \frac{\gamma N}{2}\,\mathrm{TV}(P_0, P_b)$ for all $(P_0, P_b) \in \mathcal{P}$.*

*Proof of Lemma 1.* Fix $(P_0, P_b)$ and let $\mathcal{Q} = \{\mathbf{x} \in \mathcal{X}^N : g_3(\mathbf{x}, P_0, P_b) = 1\}$ to obtain

$$1 - R(g_3; P_0, P_b) = \frac{1}{2} \sum_{j \in \{0,1\}} \Pr\{g_3(\mathcal{D}_N^{(j)}, P_0, P_b) = j\} \tag{9}$$

$$= \frac{1}{2} \int \mathbb{1}_{\mathcal{Q}}\, dP_1^N + \frac{1}{2} \int \mathbb{1}_{\mathcal{Q}^c}\, dP_0^N \tag{10}$$

$$= \frac{1}{2} + \frac{1}{2} \int \mathbb{1}_{\mathcal{Q}}\, dP_1^N - \frac{1}{2} \int \mathbb{1}_{\mathcal{Q}}\, dP_0^N \tag{11}$$

$$= \frac{1}{2} + \frac{1}{2} \int \mathbb{1}_{\mathcal{Q}}\, d(P_1^N - P_0^N) \tag{12}$$

---

[4]The following symbols use the notation from [3, Sec. 2]: $\mathcal{H}$, $X_O$, $X_I$, $Y_O$, $Y_I$, $D_{X_O}$, $D_{X_I}$, $D_{XY}$, $\mathscr{D}_{XY}$, $\ell(\cdot,\cdot)$, $K$.

$$\leq \frac{1}{2} + \frac{1}{2} \operatorname{TV}(P_0^N, P_1^N) \tag{13}$$

$$\leq \frac{1}{2} + \frac{N}{2} \operatorname{TV}(P_0, P_1) \tag{14}$$

$$\leq \frac{1}{2} + \frac{\gamma N}{2} \operatorname{TV}(P_0, P_b), \tag{15}$$

where (13) is a consequence of [14, Exercise 1.17]. Also using [14, Exercise 1.17], we see that equality in (13) is achieved for the Neyman-Pearson detector

$$g_3(\mathcal{D}_N, P_0, P_b) = \mathbb{1}\left\{\frac{dP_1^N}{dP_0^N}(\mathcal{D}_N) \geq 1\right\}. \tag{16}$$

The last two steps (14) and (15) follow from Lemma 3. $\qquad\square$

**Theorem 1.** *Fix $N \in \mathbb{N}$, $\alpha \in (0, \frac{1}{2}]$, $\beta \in [0, 1]$, and $\mathcal{P} = \{(P_0, P_b) : \operatorname{TV}(P_0, P_b) \geq 1 - \beta\}$. Let $g_2(\mathcal{D}_N, P_0)$ be an $\alpha$-error Type 2 detector. For $|\mathcal{X}| = \infty$, we then have necessarily $\alpha = \frac{1}{2}$, while for $|\mathcal{X}| < \infty$, we have*

$$N \geq \frac{\log 2\alpha}{2} + \sqrt{\frac{(\log 2\alpha)^2}{4} + (\beta|\mathcal{X}| - 1)\log\frac{1}{2\alpha}}. \tag{4}$$

*Proof of Theorem 1.* For brevity we assume $P_0$ to be given and drop it as an argument to $g_2(\mathcal{D}_N^{(j)}, P_0) = g_2(\mathcal{D}_N)$. Assume that $g_2$ is an $\alpha$-error detector. Without loss of generality, we will assume $|\mathcal{X}| = K \in \mathbb{N}$ and set $\mathcal{X} = \{1, \dots, K\}$. The case $|\mathcal{X}| = \infty$ will follow by letting $K \to \infty$.

Choose $P_0 = \mathcal{U}(\mathcal{X})$, the uniform distribution on $\mathcal{X} = \{1, \dots, K\}$. For an arbitrary, vector $\mathbf{y} = (y_1, y_2, \dots, y_M) \in \mathcal{X}^M$, let $\mathcal{Q}_{\mathbf{y}}$ be the discrete uniform distribution on the elements of $\mathbf{y}$. Note that this is only the uniform distribution on the set $\{y_m : m = 1, \dots, M\}$ if all components of $\mathbf{y}$ are different. Clearly, we have $\operatorname{TV}(P_0, \mathcal{Q}_{\mathbf{y}}) \geq 1 - \frac{M}{K}$. Thus, by choosing $M \leq \beta K$ it is ensured that $\operatorname{TV}(P_0, \mathcal{Q}_{\mathbf{y}}) \geq 1 - \beta$.

Let $\mathbf{Y} = (Y_1, Y_2, \dots Y_M)$ be a random vector with $M$ elements, each drawn i.i.d. according to $Y_m \sim P_0$. We now draw another random vector $\mathbf{Z}$ with $N$ elements $\mathbf{Z} = \{Z_n\}_{n=1,2,\dots,N}$ according to $Z_n = (1 - G_n)X_n^{(0)} + G_n Y_{V_n}$, where $V_n \sim \mathcal{U}(\{1, 2, \dots, M\})$ and $G_n \sim \mathcal{B}(\gamma)$ are all independently drawn for $n = 1, 2, \dots, N$. Thus, $V_n$ is uniformly drawn from $\{1, 2, \dots, M\}$ and $G_n$ satisfies $\Pr\{G_n = 1\} = \gamma$ and $\Pr\{G_n = 0\} = 1 - \gamma$.

We note the following two facts about this construction:

1. The marginal distribution of every $Z_n \in \mathbf{Z}$ is $P_0$, but the selection is non-i.i.d. as $Z_n$ and $Z_{n'}$ depend on each other through $\mathbf{Y}$. However, when conditioning on the fact that all components of $\mathbf{V} = (V_1, V_2, \dots, V_N)$ are pairwise distinct, then the random variables $Y_{V_n}$ and $Y_{V_n'}$ are independent for $n \neq n'$ and thus $\mathbf{Z}$ is a vector of i.i.d. variables distributed according to $P_0$.

2. When conditioning on $\mathbf{Y} = \mathbf{y}$, we have a different situation, where $Z_n \sim (1 - \gamma)P_0 + \gamma \mathcal{Q}_{\mathbf{y}}$ are i.i.d., and by choosing $M \leq \beta K$, we have $(P_0, \mathcal{Q}_{\mathbf{y}}) \in \mathcal{P}$.

Let $|\mathbf{V}| = |\{V_1, V_2, \dots, V_N\}| = N$ be the event that $\mathbf{V}$ contains pairwise distinct elements, i.e., no repetitions occur. Using the first fact above, we calculate

$$\Pr\{g_2(\mathbf{Z}) = 1\} \tag{17}$$

$$\leq \Pr\left\{g_2(\mathbf{Z}) = 1 \Big| |\mathbf{V}| = N\right\} + \Pr\{|\mathbf{V}| \neq N\} \tag{18}$$

$$\leq \Pr\left\{g_2(\mathbf{Z}) = 1 \Big| |\mathbf{V}| = N\right\} + 1 - \frac{M!}{M^N(M - N)!} \tag{19}$$

$$\leq \Pr\left\{g_2(\mathbf{Z}) = 1 \Big| |\mathbf{V}| = N\right\} + 1 - \left(1 - \frac{N}{M}\right)^N \tag{20}$$

$$= \Pr\left\{g_2(\mathcal{D}_N^{(0)}) = 1\right\} + 1 - \left(1 - \frac{N}{M}\right)^N \tag{21}$$

$$= 2 - \Pr\{g_2(\mathcal{D}_N^{(0)}) = 0\} - \left(1 - \frac{N}{M}\right)^N \tag{22}$$

$$\leq 2 - \Pr\{g_2(\mathcal{D}_N^{(0)}) = 0\} - \exp\frac{-N^2}{M-N}, \tag{23}$$

where we used the union bound as well as the inequality $\log(1+x) \geq \frac{x}{1+x}$.

Using the second fact from above, we condition on $\mathbf{Y} = \mathbf{y}$ and then have $\mathbf{Z}$ i.i.d. according to $P_1 = (1-\gamma)P_0 + \gamma P_b$ for a valid backdoor distribution $P_b = \mathcal{Q}_{\mathbf{y}}$. We then write

$$\frac{1}{2}\Pr\{g_2(\mathcal{D}_N^{(0)}) = 0\} + \frac{1}{2}\Pr\{g_2(\mathbf{Z}) = 1\} \tag{24}$$

$$= \frac{1}{2}\Pr\{g_2(\mathcal{D}_N^{(0)}) = 0\} + \frac{1}{2}K^{-M}\sum_{\mathbf{y}\in\mathcal{X}^M}\Pr\left\{g_2(\mathbf{Z}) = 1\big|\mathbf{Y} = \mathbf{y}\right\} \tag{25}$$

$$= K^{-M}\sum_{\mathbf{y}\in\mathcal{X}^M}\left(\frac{1}{2}\Pr\left\{g_2(\mathcal{D}_N^{(0)}) = 0\right\} + \frac{1}{2}\Pr\left\{g_2(\mathbf{Z}) = 1\big|\mathbf{Y} = \mathbf{y}\right\}\right) \tag{26}$$

$$= K^{-M}\sum_{\mathbf{y}\in\mathcal{X}^M}\left(\frac{1}{2}\Pr\left\{g_2(\mathcal{D}_N^{(0)}) = 0\right\} + \frac{1}{2}\Pr\left\{g_2(\mathcal{D}_N^{(1)}) = 1\big|\mathbf{Y} = \mathbf{y}\right\}\right) \tag{27}$$

$$\geq K^{-M}\sum_{\mathbf{y}\in\mathcal{X}^M}(1-\alpha) \tag{28}$$

$$= 1 - \alpha. \tag{29}$$

In total we have

$$1 - \alpha \overset{(29)}{\leq} \frac{1}{2}\Pr\{g_2(\mathcal{D}_N^{(0)}) = 0\} + \frac{1}{2}\Pr\{g_2(\mathbf{Z}) = 1\} \tag{30}$$

$$\overset{(23)}{\leq} \frac{1}{2}\left(\Pr\{g_2(\mathcal{D}_N^{(0)}) = 0\} + 2 - \Pr\{g_2(\mathcal{D}_N^{(0)}) = 0\} - \exp\frac{-N^2}{M-N}\right) \tag{31}$$

$$= 1 - \frac{1}{2}\exp\frac{-N^2}{M-N} \tag{32}$$

and thus

$$\alpha \geq \frac{1}{2}\exp\frac{-N^2}{M-N}. \tag{33}$$

This already resolves the case $|\mathcal{X}| = \infty$ as we can then let $K \to \infty$ and $M = \lfloor\beta K\rfloor \to \infty$, showing that $\alpha = \frac{1}{2}$ for $|\mathcal{X}| = \infty$.

On the other hand, for $|\mathcal{X}| < \infty$, we choose $K = |\mathcal{X}|$, $M = \lfloor\beta K\rfloor$ and obtain (4) by

$$\alpha \geq \frac{1}{2}\exp\frac{-N^2}{M-N} \tag{34}$$

$$-\log 2\alpha \leq \frac{N^2}{\lfloor\beta K\rfloor - N} \tag{35}$$

$$0 \leq N^2 - N\log 2\alpha + \lfloor\beta K\rfloor\log 2\alpha \tag{36}$$

$$N \geq \frac{\log 2\alpha}{2} + \sqrt{\frac{(\log 2\alpha)^2}{4} - \lfloor\beta K\rfloor\log 2\alpha} \tag{37}$$

$$N \geq \frac{\log 2\alpha}{2} + \sqrt{\frac{(\log 2\alpha)^2}{4} + (\beta K - 1)\log\frac{1}{2\alpha}} \tag{38}$$

$$\square$$

**Theorem 2.** *Considering the backdoor detection setup of Definition 1 with $\mathcal{P} = \{(P_0, P_b) : \mathrm{TV}(P_0, P_b) \geq 1 - \beta\}$ and a finite alphabet $|\mathcal{X}| < \infty$. There exists an $\alpha$-error Type 2 detector if*

$$\alpha > 2|\mathcal{X}| \exp\left(-\frac{2N\gamma^2(1-\beta)^2}{|\mathcal{X}|^2}\right), \tag{5}$$

In the proof of this theorem, the auxiliary Lemmas 3 and 4 are used, which are provided in Appendix A.4.

*Proof of Theorem 2.* In the following we will show that the detector

$$g(\mathcal{D}_N, P_0) = \begin{cases} 1 & \mathrm{TV}(P_0, S_N) \geq \gamma\frac{1-\beta}{2} \\ 0 & \text{otherwise} \end{cases} \tag{39}$$

is $\alpha$-error if (5) is satisfied. Here, the distribution $S_N$ is the so-called *type* of $\mathcal{D}_N$, i.e.,

$$S_N(x) = \frac{1}{N} \sum_{n=1}^{N} \mathbb{1}_x(X_i), \tag{40}$$

where for any $x \in \mathcal{X}$, $\mathbb{1}_x(X_n)$ is the indicator function that takes value 1 if $X_n = x$ and 0 otherwise.

In Lemma 4 it is shown that the type $S_N$ is close to the true distribution $P$ with high probability. We can now analyze the error probability of the detector (39) for $P = P_1$, i.e.,

$$\Pr\{g(\mathcal{D}_N^{(1)}, P_0) = 0\} = \Pr\left\{\mathrm{TV}(S_N^{(1)}, P_0) \leq \gamma\frac{1-\beta}{2}\right\} \tag{41}$$

$$\leq \Pr\left\{\mathrm{TV}(S_N^{(1)}, P_0) \leq \frac{\mathrm{TV}(P_0, P_1)}{2}\right\} \tag{42}$$

$$\leq \Pr\left\{\mathrm{TV}(S_N^{(1)}, P_1) \geq \frac{\mathrm{TV}(P_0, P_1)}{2}\right\} \tag{43}$$

$$\leq \Pr\left\{\mathrm{TV}(S_N^{(1)}, P_1) \geq \gamma\frac{1-\beta}{2}\right\} \tag{44}$$

$$\leq 2|\mathcal{X}| \exp\left(-\frac{2N\gamma^2(1-\beta)^2}{|\mathcal{X}|^2}\right), \tag{45}$$

where we used Lemma 4 in (45) and the fact that $\mathrm{TV}(P_0, P_1) = \gamma \mathrm{TV}(P_0, P_b) \geq (1 - \beta)\gamma$ by Lemma 3 in (42) and (44). Similarly, we obtain that the error probability for $j = 0$ is upper bounded by the same expression

$$\Pr\{g(\mathcal{D}_N^{(0)}, P_0) = 1\} = \Pr\left\{\mathrm{TV}(S_N^{(0)}, P_0) \geq \gamma\frac{1-\beta}{2}\right\} \tag{46}$$

$$\leq 2|\mathcal{X}| \exp\left(-\frac{2N\gamma^2(1-\beta)^2}{|\mathcal{X}|^2}\right), \tag{47}$$

applying Lemma 4 in (47).

Thus, we have shown that $g$, as defined in (39), is $\alpha$-error, provided that (5) holds. $\qquad\square$

**Theorem 3.** *Consider the setup of Definition 1, with fixed $\gamma \in (0, 1]$, $N \in \mathbb{N}$ and $\mathcal{P}$. Let $\mathcal{P}'$ be the set of $N$-fold products of $(P_0, P_1)$, i.e., $\mathcal{P}' = \{(P_0^N, (\gamma P_b + (1-\gamma)P_0)^N) : (P_0, P_b) \in \mathcal{P}\}$. Then, OOD-detection is PAC-learnable on $\mathcal{P}'$ if and only if the following holds for any $\epsilon > 0$ and any Type 3 detector $g_3(\mathcal{D}_N, P_0, P_b)$: We can find $M \in \mathbb{N}$ and a Type 1 detector $g_1(\mathcal{D}_N, \mathcal{D}'_M)$, which satisfies $R(g_1, P_0, P_b) \leq R(g_3, P_0, P_b) + \epsilon$ for every $(P_0, P_b) \in \mathcal{P}$.*

In the proof of this theorem, the auxiliary Lemma 5 is used, which is provided in Appendix A.4

*Proof of Theorem 3.* Assume first that OOD-detection is PAC-learnable on $\mathcal{P}'$, fix $\epsilon > 0$ and let $g_3$ be any Type 3 detector. By Lemma 5, we know that there is a Type 1 detector $g_1^M$ with some $M$ such that $\epsilon(M) \leq \epsilon$, satisfying (85). Noting that $\frac{1}{2} - \frac{1}{2} \mathrm{TV}(P_0^N, P_1^N) \leq R(g_3, P_0, P_b)$ by Lemma 1 completes this part of the proof.

On the other hand, let $g_3$ be the Type 3 Neyman-Pearson detector that satisfies $R(g_3, P_0, P_b) = \frac{1}{2} - \frac{1}{2} \mathrm{TV}(P_0^N, P_1^N)$, which exists by Lemma 1. By our assumptions, for any $k \in \mathbb{N}$ we can find a Type 1 detector $\hat{g}_1^k$ with $M = M(k)$ satisfying

$$\frac{1}{2} \mathrm{TV}(P_0^N, P_1^N) - \frac{1}{2} + R(\hat{g}_1^k, P_0, P_b) \leq \frac{1}{k}. \tag{48}$$

We can find a monotonically increasing sequence $k_m$ for $m = 1, 2, \ldots$ with $\lim_{m \to \infty} k_m = \infty$, that satisfies $M(k_m) \leq m$. Using the sequence of Type 1 detectors[5] $g_1^m(\mathcal{D}_N, \mathcal{D}_m') = \hat{g}_1^{k_m}(\mathcal{D}_N, [\mathcal{D}_m']_1^{M(k_m)})$ and $\epsilon(m) = \frac{1}{k_m}$, we have for every $(P_0, P_b) \in \mathcal{P}$,

$$\epsilon(m) \geq \frac{1}{2} \mathrm{TV}(P_0^N, P_1^N) - \frac{1}{2} + R(\hat{g}_1^{k_m}, P_0, P_b) \tag{49}$$

$$= \frac{1}{2} \mathrm{TV}(P_0^N, P_1^N) - \frac{1}{2} + R(g_1^m, P_0, P_b). \tag{50}$$

This completes the proof as $\lim_{m \to \infty} \epsilon(m) = \lim_{m \to \infty} \frac{1}{k_m} = 0$ and thus, PAC learnability is guaranteed by Lemma 5. $\qquad\square$

**Corollary 2.** *Let* $g_2'(\mathcal{D}_{N'}^{(J)}, P_0, X^{(I)})$ *be a Type 2 detector for an IBD problem with* $r = \min\{P_{JI}(0,0), P_{JI}(1,1)\} > 0$ *and* $\mathcal{P} = \{(P_0, P_b) : \mathrm{TV}(P_0, P_b) \geq 1 - \beta\}$. *Then, if* $g_2'$ *is* $\alpha$-*error, we have* $\alpha \geq r$ *if* $|\mathcal{X}| = \infty$, *and for* $|\mathcal{X}| < \infty$, *we obtain*

$$N \geq \frac{\log \frac{\alpha}{r}}{2} + \sqrt{\frac{(\log \frac{\alpha}{r})^2}{4} + (\beta|\mathcal{X}| - 1) \log \frac{r}{\alpha}}. \tag{8}$$

*Proof of Corollary 2.* Assuming that this detector is $\alpha$-error implies

$$\alpha \geq R(g_2', P_0, P_b) \geq P_{JI}(0,0) \Pr\{g_2'(\mathbf{Q}_2') \neq 0 | J = I = 0\}$$
$$+ P_{JI}(1,1) \Pr\{g_2'(\mathbf{Q}_2') \neq 1 | J = I = 1\} \tag{51}$$
$$\geq r\left(\Pr\{g_2'(\mathbf{Q}_2') \neq 0 | J = I = 0\} + \Pr\{g_2'(\mathbf{Q}_2') \neq 1 | J = I = 1\}\right). \tag{52}$$

Now consider the MBD problem with $\gamma = 1$ and the training set size $N' = N + 1$. We can define a Type 2 detector[6] $g_2(\mathcal{D}_{N'}, P_0) = g_2'(\mathcal{D}_N, P_0, X_{N'})$ with risk

$$R(g_2, P_0, P_b) = \frac{1}{2} \Pr\{g_2'(\mathbf{Q}_2') \neq 0 | J = I = 0\} + \frac{1}{2} \Pr\{g_2'(\mathbf{Q}_2') \neq 1 | J = I = 1\} \tag{53}$$

$$\leq \frac{1}{2r} \alpha. \tag{54}$$

From Theorem 1, we now know that $\frac{1}{2r} \alpha \geq \frac{1}{2}$ if $\mathcal{X} = \mathbb{N}$ and obtain (8) for $|\mathcal{X}| < \infty$.

$\qquad\square$

## A.4  Auxiliary Results

This appendix contains auxiliary results, which are utilized in the proofs provided in Appendix A.3.

**Lemma 3** (Properties of Total Variation). *The total variation between two probability distributions* $P_0, P_1 \in \mathcal{P}(\mathcal{X})$, *is given by*

$$\mathrm{TV}(P_0, P_1) = \|P_0 - P_1\|_{\mathrm{TV}} := \sup_A |P_0(A) - P_1(A)|, \tag{55}$$

---

[5] We use the notation $[\mathbf{x}]_k^l = [(x_1, x_2, \ldots, x_N)]_k^l = (x_k, x_{k+1}, \ldots, x_l)$ for slicing.

[6] If $\gamma > 0$ for the IBD problem, randomly replace elements of $\mathcal{D}_N$ by independently drawn realizations of $P_0$.

*where the supremum is over all measurable sets $A \subseteq \mathcal{X}$. We then have*

$$\|P_0 - P_1\|_{\mathrm{TV}} = 2 \inf_{X_0, X_1 : P_{X_0} = P_0, P_{X_1} = P_1} \Pr\{X_0 \neq X_1\}, \tag{56}$$

*where the infimum is over all random variables $X_0, X_1$ on $\mathcal{X}$, such that the marginal distributions satisfy $P_{X_0} = P_0$, $P_{X_1} = P_1$. For $P_0', P_1' \in \mathcal{P}(\mathcal{Y})$, we have*

$$\|P_0 - P_1\|_{\mathrm{TV}} \leq \|P_0 \times P_0' - P_1 \times P_1'\|_{\mathrm{TV}} \leq \|P_0 - P_1\|_{\mathrm{TV}} + \|P_0' - P_1'\|_{\mathrm{TV}}. \tag{57}$$

*and thus $\|P_0 - P_1\|_{\mathrm{TV}} \leq \|P_0^N - P_1^N\|_{\mathrm{TV}} \leq N\|P_0 - P_1\|_{\mathrm{TV}}$. Furthermore, for $\gamma \in [0,1]$,*

$$\|P_0 - (1-\gamma)P_0 - \gamma P_1\|_{\mathrm{TV}} = \gamma\|P_0 - P_1\|_{\mathrm{TV}} \tag{58}$$

*Proof.* The characterization (56) can be found in [14].

To show the first inequality in (57), observe that

$$\|P_0 \times P_0' - P_1 \times P_1'\|_{\mathrm{TV}} = \sup_B |[P_0 \times P_0'](B) - [P_1 \times P_1'](B)| \tag{59}$$

$$\geq \sup_A |[P_0 \times P_0'](A \times \mathcal{Y}) - [P_1 \times P_1'](A \times \mathcal{Y})| \tag{60}$$

$$= \|P_0 - P_1\|_{\mathrm{TV}}. \tag{61}$$

To show the second inequality in (57), we use (56) and for an arbitrary $\varepsilon > 0$, choose $(X_0, X_1) \perp (Y_0, Y_1)$ such that $P_{X_0} = P_0$, $P_{X_1} = P_1$, $P_{Y_0} = P_0'$, $P_{Y_1} = P_1'$, and

$$\|P_0 - P_1\|_{\mathrm{TV}} + \varepsilon \geq 2\Pr\{X_0 \neq X_1\}, \tag{62}$$

$$\|P_0' - P_1'\|_{\mathrm{TV}} + \varepsilon \geq 2\Pr\{Y_0 \neq Y_1\}. \tag{63}$$

Clearly $P_{X_0, Y_0} = P_0 \times P_0'$ as well as $P_{X_1, Y_1} = P_1 \times P_1'$ and thus by (56),

$$\|P_0 \times P_0' - P_1 \times P_1'\|_{\mathrm{TV}} \leq 2\Pr\{(X_0, Y_0) \neq (X_1, Y_1)\} \tag{64}$$

$$\leq 2\Pr\{X_0 \neq X_1\} + 2\Pr\{Y_0 \neq Y_1\} \tag{65}$$

$$\leq \|P_0 - P_1\|_{\mathrm{TV}} + \|P_0' - P_1'\|_{\mathrm{TV}} + 2\varepsilon. \tag{66}$$

As $\varepsilon > 0$ was arbitrary, this proves (57).

To show (58), we use (55) and have

$$\|P_0 - (1-\gamma)P_0 - \gamma P_1\|_{\mathrm{TV}} = \sup_A |P_0(A) - (1-\gamma)P_0(A) - \gamma P_1(A)| \tag{67}$$

$$= \sup_A |\gamma P_0(A) - \gamma P_1(A)| \tag{68}$$

$$= \gamma\|P_0 - P_1\|_{\mathrm{TV}}. \tag{69}$$

$\square$

**Lemma 4.** *Let $S_N$ be the type of $\mathbf{X} = (X_1, X_2, \ldots, X_N)$, distributed according to $P^N$. For any $t \in [0,1]$, we then have the bound*

$$\Pr\{\mathrm{TV}(S_N, P) \geq t\} \leq 2|\mathcal{X}| \exp\left(-\frac{8Nt^2}{|\mathcal{X}|^2}\right). \tag{70}$$

*Proof.* By using the Hoeffding's inequality we can bound the probability of the deviation of $S_N$ from its expected value. In particular, we have that

$$\Pr\{|S_N(x) - P(x)| \geq t\} = \Pr\{|S_N(x) - \mathbb{E}[S_N(x)]| \geq t\} \tag{71}$$

$$\leq 2\exp\left(\frac{-2t^2}{\sum_{n=1}^N (\frac{1}{N} - 0)^2}\right) \tag{72}$$

$$= 2\exp\left(\frac{-2t^2}{\frac{1}{N}}\right) \tag{73}$$

$$= 2 \exp\left(-2Nt^2\right),\tag{74}$$

where we note that $\mathbb{E}[S_N(x)] = \frac{1}{N}\sum_{n=1}^{N}\mathbb{E}[\mathbb{1}_x[X_n]] = P(x)$.

The next and final step is to extend the bound to the whole alphabet $\mathcal{X}$. In order to do so, we define the event $\mathcal{A}_x = \{|S_N(x) - P(x)| \geq t\}$. We want to bound the probability of the event

$$\mathcal{A} = \bigcup_{x \in \mathcal{X}} \mathcal{A}_x = \{\exists x \in \mathcal{X} : \mathcal{A}_x\}.\tag{75}$$

By applying the union bound we obtain

$$\Pr\mathcal{A} = \Pr\left\{\bigcup_{x \in \mathcal{X}} \mathcal{A}_x\right\}\tag{76}$$

$$\leq \sum_{x \in \mathcal{X}} \Pr\{\mathcal{A}_x\}\tag{77}$$

$$\leq \sum_{x \in \mathcal{X}} 2\exp\left(-2Nt^2\right)\tag{78}$$

$$= 2|\mathcal{X}|\exp\left(-2Nt^2\right).\tag{79}$$

Let us consider the event $\mathcal{A} = \{\exists x \in \mathcal{X} : |S_N(x) - P(x)| \geq t\}$: this is the error event, i.e., the divergence between the observed samples frequency and its expected value diverges more than a given value $t > 0$ for at least one $x \in \mathcal{X}$. The complement of this event is the event that the divergence is less than $t$ for all $x \in \mathcal{X}$, i.e., the event that the observed frequency is close to the expected value for all $x \in \mathcal{X}$. This can be written as

$$\mathcal{A}^c = \{\forall x \in \mathcal{X}, \ |S_N(x) - P(x)| < t\}.\tag{80}$$

Now, $\mathcal{A}^c$ implies that

$$\sum_{x \in \mathcal{X}} |S_N(x) - P(x)| < t|\mathcal{X}|\tag{81}$$

$$\frac{1}{2}\sum_{x \in \mathcal{X}} |S_N(x) - P(x)| < \frac{1}{2}t|\mathcal{X}|\tag{82}$$

$$\mathrm{TV}(S_N, P) < t'\tag{83}$$

where $t' = \frac{1}{2}t|\mathcal{X}|$. Thus, $\Pr\mathcal{A}^c \leq \Pr\{\mathrm{TV}(S_N, P) < t'\}$ and therefore

$$\Pr\{\mathrm{TV}(S_N, P) \geq t'\} \leq \Pr\mathcal{A} \leq 2|\mathcal{X}|\exp\left(-2Nt^2\right),\tag{84}$$

where we have used (79). By writing $t$ in terms of $t'$ in (84), we obtain (70). $\qquad\square$

**Lemma 5.** *Given $\mathcal{P}$ and $N \in \mathbb{N}$ and letting $\gamma \in (0,1]$, OOD-detection is PAC-learnable on $\mathcal{P}' = \{(P_0^N, P_1^N) : (P_0, P_b) \in \mathcal{P}\}$ with $P_1 = (1-\gamma)P_0 + \gamma P_b$ if and only if the following holds: For the MBD problem, there exists a sequence of Type 1 backdoor detectors $g_1^M(\mathcal{D}_N, \mathcal{D}_M')$ for $M = 1, 2, \ldots$ and a decreasing sequence $\epsilon(m)$ with $\lim_{m \to \infty} \epsilon(m) = 0$ such that for any $M \in \mathbb{N}$ and any pair $(P_0, P_b) \in \mathcal{P}$, we have*

$$\frac{1}{2}\mathrm{TV}(P_0^N, P_1^N) - \frac{1}{2} + R(g_1^M, P_0, P_b) \leq \epsilon(M).\tag{85}$$

*Proof.* Assume that OOD-detection is PAC-learnable on $\mathcal{P}'$. By definition we have a function $\mathcal{G}: \bigcup_{m=1}^{\infty} \mathcal{X}^{Nm} \to \{0,1\}^{\mathcal{X}^N}$ and a monotonically decreasing sequence $\epsilon'(m)$ that tends to zero and satisfies for every $(P_0, P_b) \in \mathcal{P}$, $m \in \mathbb{N}$, that

$$\mathbb{E}[\bar{R}(\mathcal{G}(\mathcal{D}_{mN}'), P_0^N, P_1^N)] - \inf_f \bar{R}(f, P_0^N, P_1^N) \leq \epsilon'(m),\tag{86}$$

where the infimum is over all functions $f: \mathcal{X}^N \to \{0,1\}$.

For any $M \in \mathbb{N}$, we define[7] $g_1^M(\mathcal{D}_N, \mathcal{D}'_M) := \mathcal{G}([\mathcal{D}'_M]_1^{mN})(\mathcal{D}_N)$ as well as $\epsilon(M) = \epsilon'(m)$, where $m$ is the largest integer such that $mN \leq M$. Notice that $R(g_1^M, P_0, P_b) = \mathbb{E}[\bar{R}(\mathcal{G}(\mathcal{D}'_{mN}), P_0^N, P_1^N)]$ and that $\inf_f \bar{R}(f, P_0^N, P_1^N) = \frac{1}{2} - \frac{1}{2} \mathrm{TV}(P_0^N, P_1^N)$ by Lemma 1. We thus obtain from (86), that for any $M \in \mathbb{N}$,

$$\epsilon(M) = \epsilon'(m) \geq \mathbb{E}[\bar{R}(\mathcal{G}(\mathcal{D}'_{mN}), P_0^N, P_1^N)] - \frac{1}{2} + \frac{1}{2} \mathrm{TV}(P_0^N, P_1^N) \tag{87}$$

$$= R(g_1^M, P_0, P_b) - \frac{1}{2} + \frac{1}{2} \mathrm{TV}(P_0^N, P_1^N). \tag{88}$$

Noting that $\epsilon(M)$ approaches zero completes this part of the proof.

On the other hand, assume that $g_1^M(\mathcal{D}_N, \mathcal{D}'_M)$ and $\epsilon(M)$ satisfy the requirement (85). For any $m \in \mathbb{N}$, we can then define $\mathcal{G}(\mathcal{D}'_{mN})(\mathcal{D}_N) := g_{mN}(\mathcal{D}_N, \mathcal{D}'_{mN})$ and $\epsilon'(m) = \epsilon(mN)$. We can now rewrite (85) using $\mathbb{E}[\bar{R}(\mathcal{G}(\mathcal{D}'_{mN}), P_0^N, P_1^N)] = R(g_1^M, P_0, P_b)$ and Lemma 1 to obtain

$$\epsilon'(m) = \epsilon(mN) \geq \frac{1}{2} \mathrm{TV}(P_0^N, P_1^N) - \frac{1}{2} + R(g_1^M, P_0, P_b) \tag{89}$$

$$= \mathbb{E}[\bar{R}(\mathcal{G}(\mathcal{D}'_{mN}), P_0^N, P_b^N)] - \inf_f \bar{R}(f, P_0^N, P_1^N). \tag{90}$$

Thus, we have shown that the algorithm $\mathcal{G}$ and the sequence $\epsilon'$ satisfy Definition 2 and OOD-detection is PAC-learnable on $\mathcal{P}'$. $\qquad\square$

---

[7]We use the notation $[\mathbf{x}]_k^l = [(x_1, x_2, \ldots, x_N)]_k^l = (x_k, x_{k+1}, \ldots, x_l)$ for slicing.

