# OpenReview forum: "On the Limitation of Backdoor Detection Methods"
_NeurIPS.cc/2023/Workshop/BUGS — NeurIPS 2023 BUGS Poster_

### Official Review · Reviewer_11w9 · 2023-10-25
**On the Limitation of Backdoor Detection Methods**

**Rating:** 7
**Confidence:** 3

**Review:**

- The paper focuses on the significant issue of backdoor attacks in machine learning, a topic of increasing concern given the widespread adoption of AI and machine learning models in various domains.
- The paper offers a robust literature review, showcasing a deep understanding of prior works in the field.
- The findings have significant implications for both attackers and defenders in the realm of machine learning security.
- However, this paper focuses on the theoretical analysis. The author had better evaluate their findings in real-world defenses, which can improve the soundness of this paper.

---

### Official Review · Reviewer_EKLk · 2023-10-27

**Rating:** 1
**Confidence:** 2

**Review:**

The paper claims to show a no-free-lunch theorem, proving that universal backdoor detection is impossible, except for very small alphabet sizes.

Unfortunately, it is hard for me to review. The paper constantly refers to the notation "TV", but never formally defines or clarifies what it means. (I am not an expert in theory; thus, I may miss it simply due to my own ignorance.)

Also, I think the claim of this paper heavily depends on the definition of $P_b$ (the distribution of the backdoor sample). In my opinion, when $P_b$ is too close to $P_0$, maybe the backdoor attack does not make sense at all. There should be some condition about which distribution of $P_b$ will lead to a successful and meaningful backdoor attack. I don't see the formulations in this paper address these critical issues. Otherwise, one can always say it's impossible to separate $P_b$ and $P_0$ if $P_b$ and $P_0$ are very close... But, in that case, maybe it is not a backdoor attack at all.

In summary, due to the obscure nature of the formulations, I can not accurately review this paper. I suggest ACs refer more to other reviewers's opinions.

---

### Decision · Program_Chairs · 2023-10-28

**Decision:**

Accept (Poster)

**Comment:**

Thank you authors for your work, while the reviewers' decisions were split it seems like this work could have a beneficial impact on the backdoor community and for the workshop. Looking forward to your poster!